# Graphene-edge dielectrophoretic tweezers for trapping of biomolecules

Avijit Barik[1,2], Yao Zhang[1,3], Roberto Grassi[1], Binoy Paulose Nadappuram[4], Joshua B. Edel [4], Tony Low[1], Steven J. Koester[1] & Sang-Hyun Oh[1]

The many unique properties of graphene, such as the tunable optical, electrical, and plasmonic response make it ideally suited for applications such as biosensing. As with other surface-based biosensors, however, the performance is limited by the diffusive transport of target molecules to the surface. Here we show that atomically sharp edges of monolayer graphene can generate singular electrical field gradients for trapping biomolecules via dielectrophoresis. Graphene-edge dielectrophoresis pushes the physical limit of gradient-force-based trapping by creating atomically sharp tweezers. We have fabricated locally backgated devices with an 8-nm-thick $HfO_2$ dielectric layer and chemical-vapor-deposited graphene to generate 10× higher gradient forces as compared to metal electrodes. We further demonstrate near-100% position-controlled particle trapping at voltages as low as 0.45 V with nanodiamonds, nanobeads, and DNA from bulk solution within seconds. This trapping scheme can be seamlessly integrated with sensors utilizing graphene as well as other two-dimensional materials.

[1] Department of Electrical and Computer Engineering, University of Minnesota, Minneapolis, MN 55455, USA. [2] Department of Biomedical Engineering, University of Minnesota, Minneapolis, MN 55455, USA. [3] Department of Chemistry, University of Minnesota, Minneapolis, MN 55455, USA. [4] Department of Chemistry, Imperial College London, South Kensington, London SW7 2AZ, UK. Correspondence and requests for materials should be addressed to S.J.K. (email: skoester@umn.edu) or to S.-H.O. (email: sang@umn.edu)

Graphene[1] is an excellent alternative to noble metals for constructing a wide range of sensors due to its electrical tunability[2], high quantum efficiency for light-matter interactions[3], quantum capacitance effects[4, 5], and tightly confined mid-infrared plasmons[6–10]. Unlike noble metals, such as gold or silver, the carrier concentration in graphene can can be tuned, hence enabling the possibility of electrically reconfigurable biosensing[11]. Nanopatterned graphene such as ribbons[11], nanochannels[12], or nanopores have all been shown to offer benefits for use in sensing applications. However, biomolecules in viscous media is generally governed by diffusive transport, and the placement of target molecules at the region of highest sensitivity is a key prerequisite to biosensing. Currently, graphene-based biosensors mostly employ DNA or protein immobilization across the entire graphene surface or rely on the random adsorption of biomolecules to the most sensitive regions–the edges or pores in graphene[11, 13]. The ability to precisely position and concentrate target molecules onto the edge of patterned graphene nanostructures is highly desirable yet is not extensively studied. Besides biosensing, such capability can also benefit nanophotonic applications for integrating quantum emitters and plasmonic antennas with tunable optoelectronic properties of graphene[14, 15].

Various on-chip manipulation techniques, such as optical trapping[16–18], optoelectronic tweezers[19, 20], flow-through nanopores[21, 22], electrokinetics[23], dielectrophoresis (DEP)[24], or photothermal methods[25] can be used for the aforementioned purposes. However, it is not trivial to integrate such schemes on patterned graphene chips. We show that the atomic-scale thickness of graphene enables ultra-strong DEP forces for trapping nanoscale objects and molecules along patterned edges of graphene. DEP is a widely used method to manipulate biomolecules or polarizable nanoscale objects by using gradient electrical forces obtained from sharp conducting tips, edges, or small gaps between electrodes[24]. The time-averaged DEP force on a particle of radius $R$ and in a medium of permittivity $\varepsilon_m$ is expressed as[24]

$$\vec{F}_{DEP}(\omega) = \pi \varepsilon_m R^3 \cdot \mathrm{Re}\left( \frac{\varepsilon_p^*(\omega) - \varepsilon_m^*(\omega)}{\varepsilon_p^*(\omega) + 2\varepsilon_m^*(\omega)} \right) \nabla |\mathbf{E}|^2 \qquad (1)$$

where $|\mathbf{E}|$ is the magnitude of the electric field, $\varepsilon_p^*(\omega)$ and $\varepsilon_m^*(\omega)$ are the complex permittivities of the particle and the medium, respectively. For the case where electric field gradient varies greatly over particle dimensions, higher order moments (quadrupole, octopole, and so on) become important[26]. However, in this case as our model system is based on nanoscale particles, we assumed a dipole approximation.

The DEP force depends on the particle volume that goes down with the particle size. However, the thermal Brownian motion becomes very important as we reduce the particle size to sub-micron length scale. To trap a particle, it is required to overcome

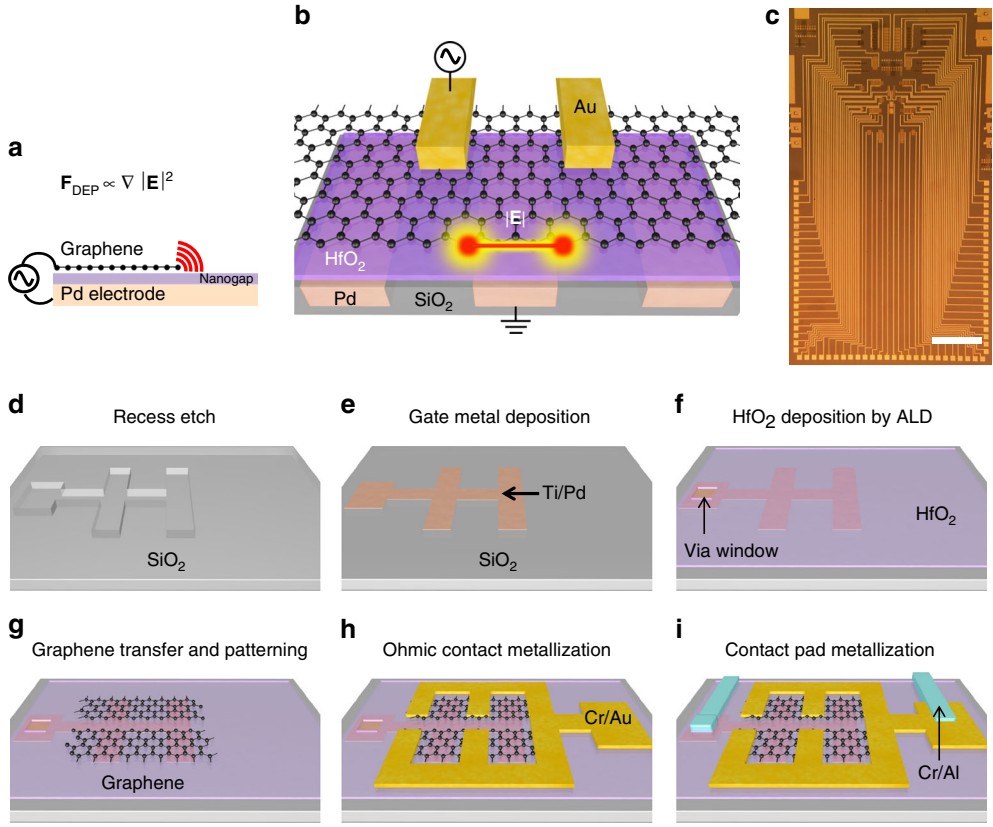

**Fig. 1** Graphene as an electrode for dielectrophoresis. **a** Being a gradient-dependent phenomenon, DEP force is increased as the radius of curvature of an electrode is reduced. The edge of graphene could provide the smallest possible radius of curvature. **b** An illustration showing the region of strongest electric field gradient is generated at the intersection of the edge of the graphene by applying an AC bias between the graphene contact electrode (gold) and palladium gate electrode. **c** A photograph of the chip. Scale bar: 2 mm. **d** The gate electrode patterns were created on SiO₂ substrate by combining photolithography with reactive ion etching and wet etching. **e** Ti/Pd layer was deposited followed by lift off to create the gate electrodes. **f** 8-nm-thick HfO₂ was deposited to coat the entire surface with an insulator. A via window was created for contact pad metallization with the Pd gate electrode at a later step. **g** Single-layer graphene grown by chemical vapor deposition process was transferred to the dielectric layer by a wet transfer process, followed by patterning using photolithography and O₂ plasma etching. **h** Ohmic contacts were made by photolithography, Cr/Au deposition by electron-beam evaporation and lift off. **i** A thick metal layer of Cr/Al was added to create low-resistance electrical leads for electrical probing

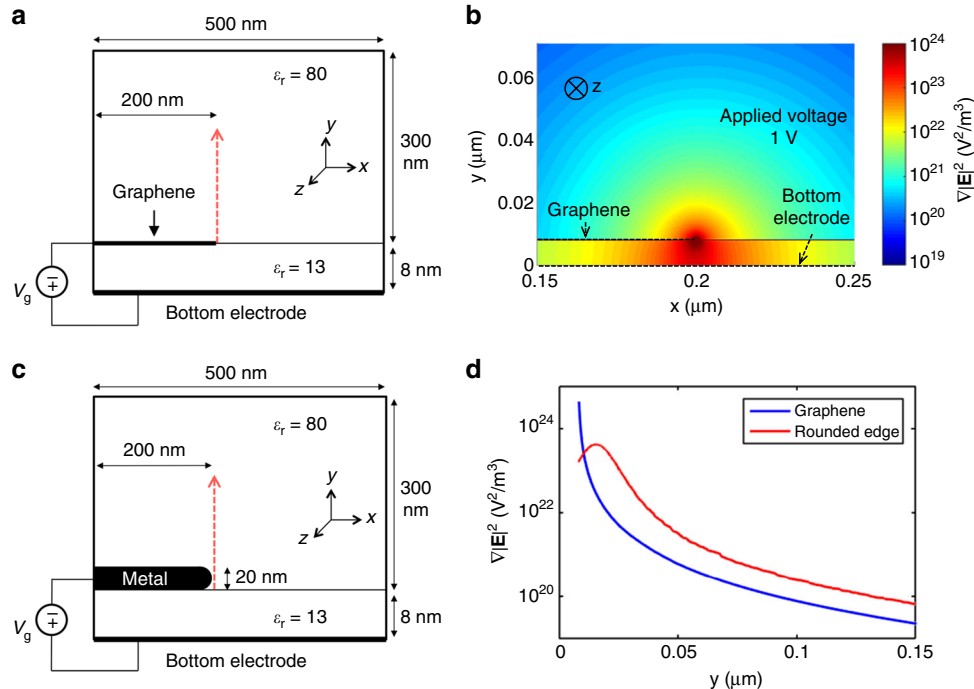

**Fig. 2** Self-consistent calculations of field gradients around a graphene edge. **a** Graphene electrode of zero thickness is placed 8 nm away from the bottom electrode and gradient of electric field intensity is calculated when a bias is applied between the electrodes. **b** Intensity plot in logarithmic scale of $\nabla|\mathbf{E}|^2$ computed at the back gate voltage $V_g = 1\,V$ (the flat band voltage is set to zero) showing a strong peak at the graphene edge. An 8-nm-thick HfO$_2$ layer with a dielectric constant of 13 was assumed between the electrodes. The rest of the simulation domain was assumed to be water with a dielectric constant of 80. **c** Vertical cut-line (noted by red arrow) of the intensity plot in **a** at $x = 0.2\,\mu m$ (position of the graphene edge), compared with the result of a similar simulation where graphene is replaced by a metal with a thickness of 20 nm and a realistic round edge with radius of curvature 10 nm. **d** The two profiles are similar but the magnitude of $\nabla|\mathbf{E}|^2$ at the edge position is greatly enhanced in the graphene case because of its one-atom thickness

the thermal force, $F_T$, given by

$$F_T = \frac{k_B T_R}{2R} \qquad (2)$$

where $T_R$ is the room temperature and $k_B$ is the Boltzmann constant.

Since the DEP force is proportional to the field gradient term $\nabla|\mathbf{E}|^2$, it is a scalable particle trapping method, i.e., reducing the radius of curvature of electrode features (e.g., tips[27]) or electrode-to-electrode separation[28] can significantly boost its magnitude. Patterned metallic nanoelectrodes[29] or bottom-up structures such as isolated carbon nanotubes (CNTs)[30, 31] have been used to shrink the critical dimension of DEP electrodes into single-digit nanometer scale. Extrapolating this idea further toward the ultimate limit, we conceive a design to turn Angstrom-scale-thickness monolayer graphene into an electrode that provides the sharpest possible edge over millimeter length scales, which cannot be created using lithographically patterned metal nanostructures or even CNTs. Although single-walled CNT electrodes can provide sharp nanometer-scale edges, these devices often suffer from inability to control the gap between CNT electrode and the gate, as well as difficulty to form high-density regular arrays of CNTs. Previously, Xie et al.[32] used an atomic force microscopy probe to cut graphene into small-area interdigitated electrodes and showed DEP trapping of CNTs over these structures.

In this work, we demonstrate a novel scheme to turn exposed edges of graphene into strong DEP trapping sites. This was done by constructing a robust large-scale graphene-edge DEP platform and trapping nanoparticles and biomolecules at the edge of this monolayer graphene atop an 8-nm-thick insulator with low bias

voltages. Such capability can offer graphene-based sensors a unique solution to diffusion-limited mass transport problems[33].

## Results

**Design and fabrication of a graphene DEP chip**. Being a gradient-field dependent phenomenon, DEP is greatly enhanced by reducing the radius of curvature of the electrode, with graphene edge providing the sharpest tip (Fig. 1a). Reducing the gap between the graphene and the ground electrode to sub-10 nm length scales aids in further enhancement of the DEP forces. A schematic of the device design is illustrated in Fig. 1b. Lithographically patterned graphene films were created as electrodes for DEP experiments. Applying an alternating current (AC) bias between the graphene contact electrode (Au) and the Pd gate electrode, generates ultra-high electric field gradients near the graphene edge that can be used to trap nanoparticles by DEP. The strength of this electric field gradient depends on the gap between the two electrodes, which in this case is 8 nm (thickness of the dielectric layer (HfO$_2$)). A photograph of the chip is shown in Fig. 1c, which was prepared by employing a multi-step fabrication process (Fig. 1d–i). First, a metal electrode (Ti/Pd) pattern was created on a Si wafer with a thick layer of SiO$_2$ on it. Next, the entire wafer surface, including the Pd electrode, was coated with 8-nm-thick HfO$_2$ deposited by atomic layer deposition (ALD). Then, single-layer graphene grown by chemical vapor deposition (CVD) was transferred onto the wafer and etched into rectangular patterns. In addition to the sharpness of the graphene edge, the nanoscale gap between graphene and the Pd gate electrode (defined by the thickness of the HfO$_2$ layer) further enhances the electric field. Finally, photolithography was employed to pattern Cr/Au contact electrodes, which formed Ohmic contacts to the graphene. The gate and the contact electrodes were arranged in

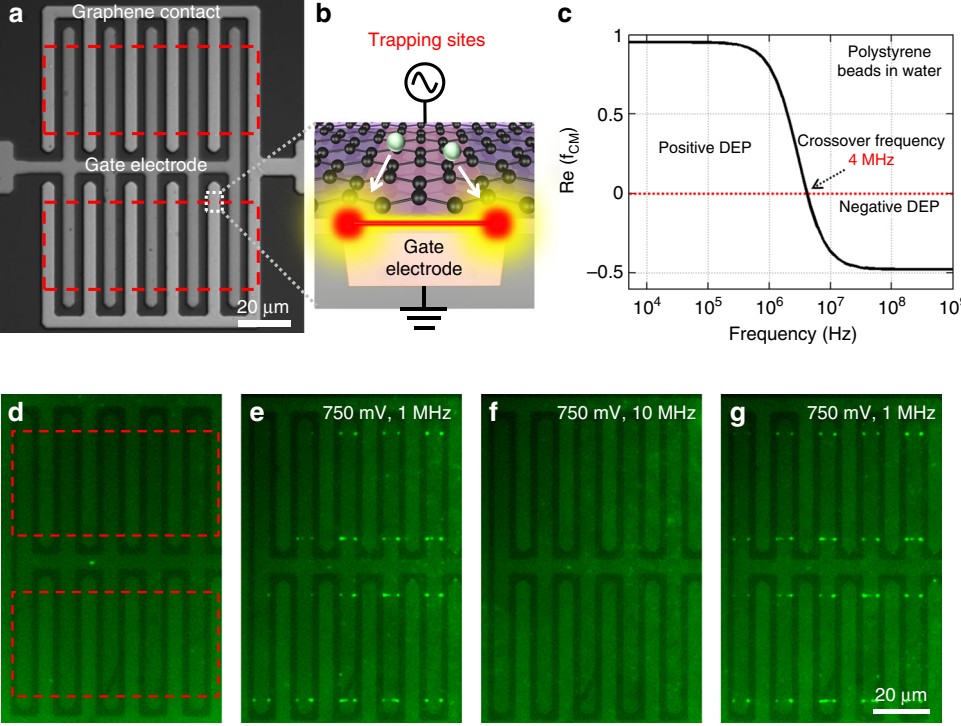

**Fig. 3** DEP manipulation of 190 nm polystyrene beads using graphene electrode. **a** A bright field microscopic image of the electrode design is shown with the location of the graphene patterns marked with red rectangles. **b** Applying an AC bias between graphene contact and the bottom gate electrode creates strong field gradients at the graphene edge. However, the region of strongest field strength resides at the intersection points of the graphene edge and the bottom electrode edge (red circles)—"trapping sites". **c** Clausius-Mossotti (CM) factor plot for polystyrene beads in water with a conductivity of 4 μS/cm. **d** A green background was observed before applying voltage from the bulk solution. **e**–**g** 190 nm polystyrene beads were trapped **e** at the trapping sites due to positive DEP (750 mV, 1 MHz). The beads can also be released **f** due to negative DEP (750 mV, 10 MHz) and then retrapped again **g** by applying positive DEP

an interdigitated fashion to minimize series resistance and to produce more exposed edges efficiently trap the nanoparticles and molecules. Details of the fabrication process are elaborated in Methods section.

The key feature of our processing scheme is the wafer-scale throughput and scalability of trap arrays. In the layout we employed, the gate finger spacing is 15 μm with two graphene segments on either side, and these segments have a pitch of 60 μm. The current design provides a density of four trapping sites per 900 μm². This density can be further enhanced by reducing the pitch of the gate fingers as well as the graphene segments to 1 μm, which is realistic using i-line optical lithography. In such a case, the trapping site density could be increased to 4 sites per 1 μm², a nearly 1000× improvement over the current devices. Furthermore, using advanced optical lithography or electron-beam lithography to form the gate fingers and segment the graphene, the trapping density could be further increased by another 1–2 orders of magnitude.

**Self-consistent calculations of graphene-edge electrostatic fields**. We calculated the gradient of the electric field intensity ($\nabla |\mathbf{E}|^2$), which is responsible for DEP trapping, for a semi-infinite graphene "electrode", electrically biased by a metal gate electrode, as shown in Fig. 2a. An 8-nm-thick HfO₂ layer with a dielectric constant of 13 is assumed between the electrodes. As the experiments were all performed in water, the rest of the simulation domain is assumed to be water with a dielectric constant of 80. The bottom electrode is treated as an ideal metal and graphene is modeled as a layer with zero thickness. An illustration of the simulation domain is shown in Fig. 2a. Here we note that graphene is not a perfect metal, due to its relatively small

electronic density-of-states. The electrostatics problem then requires a self-consistent solution of Poisson's equation and that of the finite charge density within graphene, as imposed by its Dirac-like energy dispersion:

$$\nabla \cdot (\varepsilon \nabla \varphi) = en\delta(y - y_0) \tag{3}$$

$$n(x) = \frac{2}{\pi} \left( \frac{k_B T_R}{\hbar v_F} \right)^2 \left[ \mathcal{F}_1 \left( \frac{\mu}{k_B T_R} \right) - \mathcal{F}_1 \left( \frac{-\mu}{k_B T_R} \right) \right], \quad x < 200\,\text{nm} \tag{4}$$

The symbols are defined as follows: $\varphi(x, y)$ is the 2D electrostatic potential, $n(x)$ the net electron concentration (per unit area) on the graphene layer, $\varepsilon = \varepsilon_r \varepsilon_0$ ($\varepsilon_r$ is the dielectric constant of the different media and $\varepsilon_0$ the vacuum permittivity), $\delta$ is Dirac's delta function, $y_0 = 8$ nm the vertical position of the graphene layer, $\hbar$ the reduced Planck constant, $v_F \approx 10^6$ m/s the graphene Fermi velocity, $\mathcal{F}_1$ the Fermi-Dirac integral of order 1, and $\mu(x)$ the position-dependent chemical potential. The latter, in turn, is given as:

$$\mu(x) = e\varphi(x, y = y_0), \quad x < 200\,\text{nm} \tag{5}$$

We applied the Dirichlet boundary condition $\varphi = V_g = 1$ V at the bottom edge of the simulation domain and Neumann boundary conditions everywhere else. The gradient of $|\mathbf{E}|^2$ is computed from $\varphi$ with finite differences method.

We have performed frequency-dependent capacitance measurements on metal-insulator-metal capacitors using our ALD

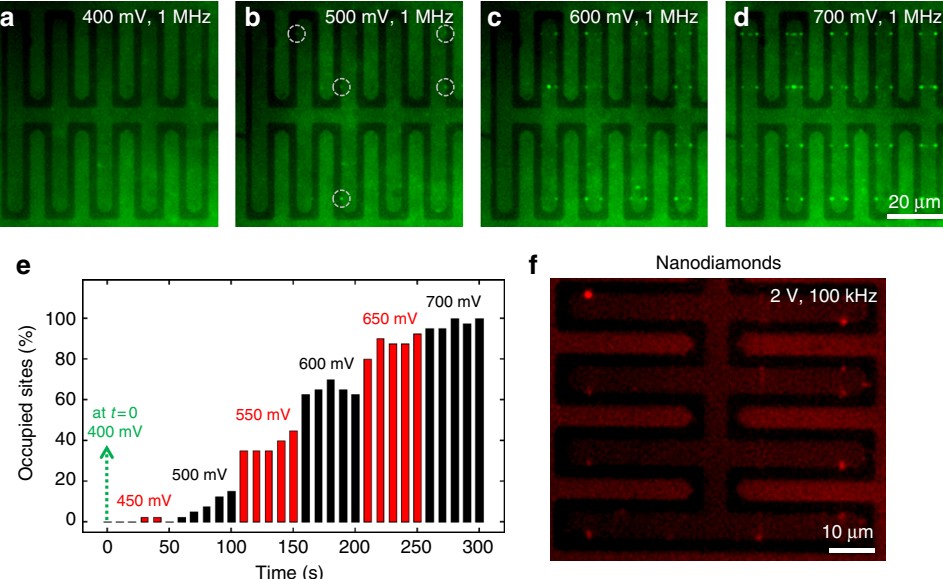

**Fig. 4** Voltage dependence and nanodiamond trapping. **a–d** Applied voltage was gradually increased at a frequency of 1 MHz and a greater number of trapped beads was observed at the trapping sites. **e** The occupancy (%) of the trapping sites was monitored as the applied bias amplitude was increased. The $V_{min}$ for 190 nm polystyrene beads was 500 mV. **f** Nanodiamond particles were trapped at the trapping sites by applying 2 V and 100 kHz frequency

HfO$_2$ dielectrics and observed only about ~10% frequency dispersion between 1 kHz and 10 MHz. These results are consistent with previous results[34] for ALD-deposited HfO$_2$, and further shows that our dielectrics do not display significant dielectric relaxation as is observed in rare-earth-doped and other more complex high-k dielectrics[35].

The $\nabla|\mathbf{E}|^2$ field gradient map is shown in Fig. 2b for a 1 V DC bias applied between the gate and graphene electrodes. The $\nabla|\mathbf{E}|^2$ value is highest at the edge of the graphene electrode, which acts as a hotspot for DEP trapping. A vertical cut-line of the $\nabla|\mathbf{E}|^2$ field gradient at the position of the graphene edge is plotted and compared to a case where the graphene electrode was replaced with a realistic metal electrode of 20 nm thickness and 10 nm radius of curvature (Fig. 2c). The maximum value of $\nabla|\mathbf{E}|^2$ at the graphene edge is about an order of magnitude higher as compared to the metal electrode (Fig. 2d). However, the effect of the graphene edge becomes less pronounced as we move away from the boundary. In the bulk solution (away from the electrode boundary), the effect of DEP is similar in both cases and mostly dependent on the gap between the electrodes (8 nm here). However, near the graphene edge, gradient forces are stronger than the metal electrode, demonstrating its capability for larger short-range trapping. This in turn enables trapping of small number of analyte nanoparticles in a more controllable fashion at the edge of the graphene electrode without much interference from the bulk solution. For protein molecules or quantum dots that are typically <10 nm in diameter, graphene electrodes could provide a stronger trapping force to hold them on to the electrode edge.

**Experimental demonstration of graphene-edge DEP trapping and repulsion**. A bright-field optical microscope image of the final device is shown in Fig. 3a, where the locations of the graphene films are noted by red rectangles. Applying an AC bias between the gate and graphene contact electrodes creates a region of strong electric field gradient near the graphene edge, with the maximum $\nabla|\mathbf{E}|^2$ at either end of the graphene electrode boundary (noted by red circles in Fig. 3b), where graphene edges intersect with buried metal electrodes running in an orthogonal direction. These point junctions act as the region of strongest trapping

potential because the $\nabla|\mathbf{E}|^2$ is enhanced by non-uniform fields of both the graphene and the gate electrode edges. A particle under the influence of DEP is driven toward these junctions, also termed as the "trapping site". However, the entire boundary of the graphene electrode may also act as a DEP trap, albeit to a lesser extent than the trapping sites. The magnitude of the DEP force depends on the sharpness of the electrode boundaries as well as the 8 nm gap between the electrodes.

Polystyrene beads, 190 nm in diameter, were used to demonstrate efficient DEP trapping and releasing with sub-1 V bias voltages. The polarity of the DEP force depends on the frequency-dependent Clausius-Mossotti (CM) factor ($f_{CM}(\omega)$) given by

$$f_{CM}(\omega) = \left( \frac{\varepsilon_p^*(\omega) - \varepsilon_m^*(\omega)}{\varepsilon_p^*(\omega) + 2\varepsilon_m^*(\omega)} \right) \tag{6}$$

Particles are attracted toward the trapping sites by positive DEP ($\mathrm{Re}(f_{CM}(\omega)) > 0$) or are repelled away by negative DEP ($\mathrm{Re}(f_{CM}(\omega)) < 0$). The frequency of the applied AC bias at which such transition takes place, $f_{CM}(\omega) = 0$, is called the crossover frequency. To predict the frequency response of the polystyrene beads in water (conductivity = 4 µS/cm), the CM factor was plotted as a function of frequency (Fig. 3c). The crossover frequency is found to be 4 MHz. Hence, a frequency of 1 MHz was chosen for positive DEP and 10 MHz for negative DEP. Fluorescently labeled polystyrene beads ($\lambda_{ex}$: 470 nm, $\lambda_{em}$: 525 nm) were used to visualize the DEP manipulation capability of the graphene electrodes using a ×50 objective. Bulk fluorescence was observed before applying a voltage, which comes from the surrounding solution containing fluorescent beads (Fig. 3d). As soon as a bias of amplitude 750 mV (single-sided) was turned on, beads were selectively trapped at the trapping sites, which provide the maximum DEP force as discussed above (Fig. 3e). Of the 40 available trapping sites (4 sites on each finger electrode), ~34 (85%) of them were seen to be occupied. The trapped particles could be released due to negative DEP, simply by switching the frequency to 10 MHz (Fig. 3f). Particles were trapped again by switching back to the positive DEP mode – showing the reversible

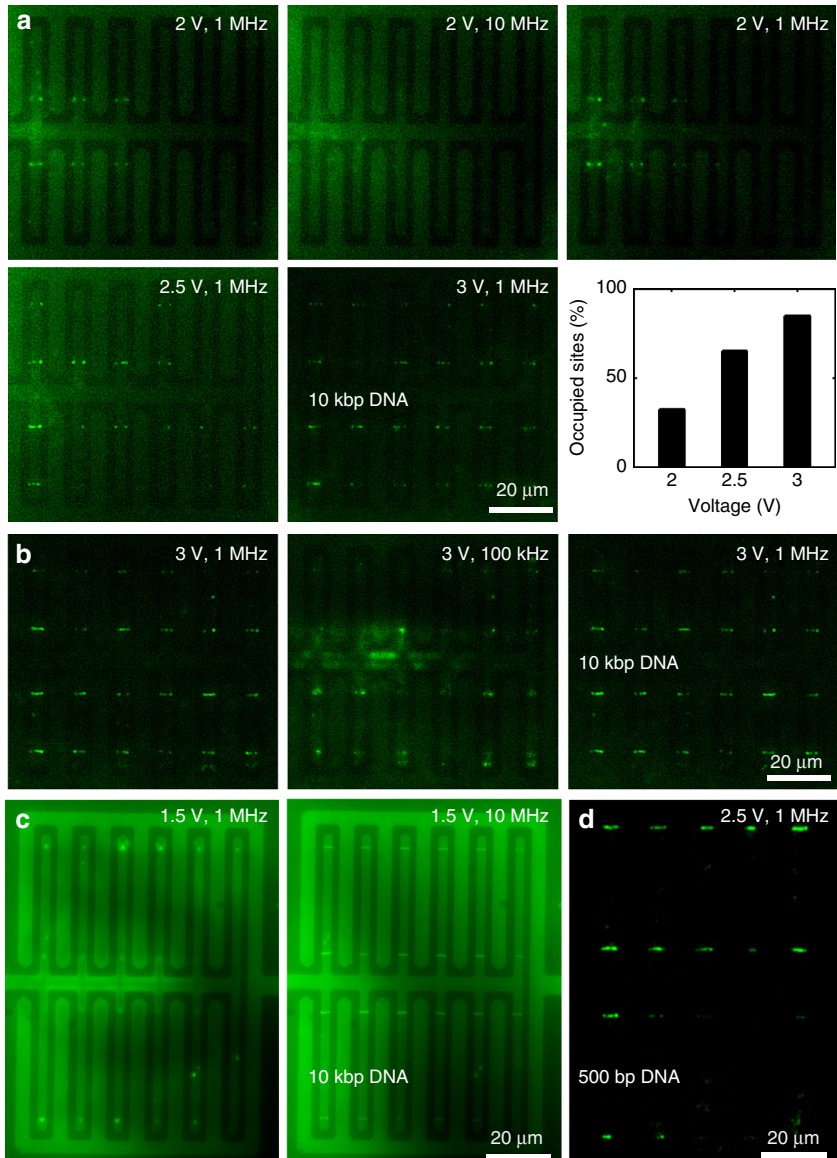

**Fig. 5** DEP manipulation of DNA molecules. **a** Trapping and releasing of 10 kbp DNA was observed at 1 MHz and 10 MHz, respectively. Increasing the voltage amplitude from 2 V to 2.5 V to 3 V, traps more DNA molecules at the trapping sites. The occupancy (%) of the trapping sites is plotted as a function of the applied voltage. **b** At a lower frequency of 100 kHz, DNA (10 pM DNA in 10 µM KCl, 12 µS/cm) localization near the graphene electrode could be observed instead of tight trapping along the edge, which can be reversed simply by switching to higher frequency. **c** At a higher solution conductivity (1 nM DNA in 1 mM KCl, 0.93 mS/cm), DNA localization vs the tight trapping phenomenon is observed at a higher frequency range. **d** 500 bp DNA molecules (threshold $\nabla|\mathbf{E}|^2$ an order of magnitude higher than that of the 10 kbp DNA molecules) were trapped along the graphene edge at 2.5 V and 1 MHz frequency

nature of graphene DEP manipulation (Fig. 3g). This time all 40 trapping sites were occupied as most of the released particles from the previous step stayed near the graphene edge–reducing the time of diffusion (Supplementary Movie 1). It should also be noted that in some finger electrodes more beads could be observed along the graphene edge (the red line in Fig. 3b), which could be due to (1) the entire boundary of the graphene electrode could potentially present a fringe electric field that is responsible for DEP and (2) random sharp protrusions along the edge that can enhance the electric field gradient and possibly act as a DEP trap.

**Low-power DEP trapping of nanobeads with graphene electrodes**. Next, we performed a voltage dependence study to determine the minimum trapping voltage ($V_{min}$) for 190 nm

polystyrene beads. This is the voltage at which the DEP force is enough to overcome the Brownian motion of the nanoparticles due to their thermal energy. The amplitude of the applied voltage was increased from 400 to 700 mV at a step interval of 50 mV. At each voltage, fluorescence images were collected for 50 s. More beads were trapped at the trapping sites as the voltage amplitude was increased (Fig. 4a–d). To quantify this further, we measured the percentage of occupied trapping sites as a function of the applied voltage (Fig. 4e). The first data point represents 400 mV, where no trapping was observed (Fig. 4a). At 500 mV, clear trapping of polystyrene beads could be observed at multiple trapping sites (circled in Fig. 4b). At higher voltages above 700 mV all trapping sites were occupied. From this study, the $V_{min}$ for 190 nm beads was found to be 500 mV, where at least one trapping site was fully occupied for the entire duration.

**Trapping nanodiamond particles on a 2D grid**. Besides potential applications in biosensing, graphene-based DEP can be utilized for on-chip assembly and array formation of nanoscale quantum emitters such as nanodiamond particles, semiconductor nanocrystals, or plasmonic metal nanoparticles. Rapid on-chip integration of these elements with patterned graphene can be highly desirable for applications in nanosensing[36] and graphene plasmonics, but is very difficult to achieve due to their nanoscale size and aggressive Brownian motion in solutions. Here we show trapping of 40 nm nanodiamond (ND) particles with nitrogen-vacancy (NV) centers ($\lambda_{em}$: 637 nm, Adamas Nanotechnologies) at the trapping sites on graphene (Fig. 4f). The ND particles are carboxylate-modified to ensure facile dispersion in aqueous solution. The ND particles polarize in presence of a non-uniform electric field and are attracted toward the region of strongest electric field gradient due to positive DEP. Fluorescent images ($\lambda_{ex}$: 540–553 nm) collected after turning on the bias (amplitude 2 V and frequency 100 kHz) show ND particles localized at the trapping sites on a two-dimensional grid defined by orthogonal edges of graphene and buried metal lines.

**DEP manipulation of DNA at pM concentrations**. To demonstrate the utility of the graphene electrodes in capturing biomolecules, we used 10 kbp and 500 bp DNA molecules tagged with YOYO-1 dye ($\lambda_{ex}$: 488 nm) at a final concentration of 10 pM in a 10 µM KCl solution. To predict the polarizability of the DNA molecules, a counterion fluctuation (CIF) model is often used, which depends on the redistributions of counterions around the charged sites present on the molecule. For this model to be valid, the concentration of counterions ($C_{ions}$) present in the solution should be much greater than the total number of charged sites present across all DNA molecules, as expressed by

$$\frac{C_{ions}}{C_{DNA}} \gg N_{ions/DNA} \qquad (7)$$

where $C_{DNA}$ is the concentration of DNA and $N_{ions/DNA}$ represents the number of counterions required to saturate all the charged sites in a DNA molecule. Since there are two negative charges per base pair on a DNA molecule, the values of $N_{ions/DNA}$ for 10 kbp and 500 bp DNA are 20,000 and 1000, respectively. As $C_{ions}/C_{DNA}$ (=$10^6$) is much higher than $N_{ions/DNA}$, the criterion in equation 7 is satisfied. The trapping force however must still be enough to overcome the thermal drag force (equation 2), which is contributed by diffusion. The DEP force on a DNA molecule can be expressed by:

$$\vec{F}_{DEP} = \frac{1}{4}\alpha\nabla|\mathbf{E}|^2 \qquad (8)$$

where $\alpha$ is the real part of the polarizability of the DNA molecule[37]. The polarizability values were estimated from previously reported empirical findings that correspond to ~$3.48 \times 10^{-31}$ Fm² and ~$1.00 \times 10^{-31}$ Fm² for the 10 kbp and 500 bp DNA, respectively[37]. Next, we calculated the threshold $\nabla|\mathbf{E}|^2$ by equating the DEP force (equation 8) with the thermal force (equation 2). The radius of gyration ($Rg$) of 10 kbp DNA with persistence length 50 nm, calculated using a worm-like chain model[38, 39], is 238 nm that corresponds to an $F_T$ value of 14.4 fN (using equation 2). The threshold $\nabla|\mathbf{E}|^2$ to trap a 10 kbp DNA molecules is ~$1.66 \times 10^{17}$ V²m⁻³ (using equation 8). For 500 bp DNA with $Rg$ of 53 nm, ($F_T \sim 64.7$ fN), the threshold $\nabla|\mathbf{E}|^2$ is ~$2.59 \times 10^{18}$ V²m⁻³.

Figure 5a shows trapping and releasing of DNA molecules at an applied voltage of amplitude 2 V. A frequency of 1 MHz was used to capture the DNA molecules at the trapping sites. The efficiency of trapping goes down at higher frequencies, as the counterions present in the solution do not have enough time to redistribute in each cycle of the AC bias. Thus at 10 MHz the DNA molecules lose their polarizability and are released into the solution. The threshold $\nabla|\mathbf{E}|^2$ to trap a 10 kbp DNA molecule is $1.66 \times 10^{17}$ V²m⁻³, which is trivial to achieve in graphene electrodes (Fig. 2d). However, a slightly higher voltage (2 V) was applied to achieve DEP manipulation, as the concentration of DNA used in this experiment was fairly low (10 pM). By applying higher voltages, it is possible to trap more number of DNA molecules. This was further demonstrated by applying higher voltage amplitudes of 2.5 V and 3 V that improved the DNA capture efficiency as evident from the increased number of occupied trapping sites. We also measured the occupancy of trapping sites as a function of the applied voltage (Fig. 5a). At 2 V, around 32.5% of the trapping sites were occupied, which was increased to 65% at 2.5 V and 85% at 3 V.

So far, we showed DEP manipulation using high frequency AC fields (1–10 MHz). Reducing the frequency to 100 kHz, results in an interesting localization of the DNA molecules near the graphene edge but the molecules are not tightly trapped at the edge (Fig. 5b). This phenomenon could be explained by the generation of a fluid flow at lower frequencies due to the formation of electrical double layers near the electrode surface. However, this mechanism depends on the charge relaxation frequency ($f$) of the system, which is given by

$$f = \frac{\sigma_m}{2\pi\varepsilon_m} \qquad (9)$$

For a solution of conductivity 12 µS/cm (10 µM KCl solution, measured by B-771 LAQUAtwin, Horiba Scientific), the charge relaxation frequency is 270 kHz (from equation 9). Hence, while operating at 100 kHz, the system has enough time to create electrical double layers, which can generate fluid flow due to electroosmosis. This effect can also be switched to the case of tight trapping along the edge of graphene, simply by switching to higher frequencies. Figure 5b shows the switching between these two effects in a reversible fashion with tight trapping at 1 MHz and localization near graphene edge at 100 kHz (Supplementary Movie 2). Increasing the solution conductivity will also increase the charge relaxation frequency. For instance, using a solution of higher conductivity with 1 nM DNA in 1 mM KCl (conductivity 0.93 mS/cm), increases the relaxation frequency to 21 MHz. In this case, we can observe a similar DNA localization effect near the graphene edge even at 1 MHz (Fig. 5c). Switching to higher frequency traps the DNA molecules in a tight fashion along the graphene edge. The increased brightness of the overall background as well as the reduced operating voltage (1.5 V) is attributed to the 100× higher concentration of the DNA molecules.

500 bp DNA was used to demonstrate trapping of smaller DNA strands (Fig. 5d). The threshold $\nabla|\mathbf{E}|^2$ to capture a 500 bp DNA molecule is $2.59 \times 10^{18}$ V²m⁻³, which is an order of magnitude higher than the threshold $\nabla|\mathbf{E}|^2$ required to trap 10 kbp DNA molecules. From Fig. 2d we estimate that theoretically a 500 bp DNA molecule can be captured at a distance of 250 nm away from the graphene edge while applying an AC bias of 1.4 V amplitude. However, here we chose to work with a bias of amplitude 2.5 V to increase the range of trapping. Also, as 500 bp DNA molecules have an $Rg$ of 53 nm, it was hard to observe single DNA molecule trapping at the trapping sites. Hence, we waited until trapping was observed along the entire edge of graphene electrode, which could be achieved within one minute.

Stability of biomolecules is dependent on the temperature of the surrounding environment. We investigated the expected temperature rise in our system based on Joule heating from the

relation $\Delta T_S \sim \sigma_m V^2/2k$ (where $\Delta T_S$ is the rise in solution temperature and $k$ is the thermal conductivity). As we used low voltages and low-conductivity solutions in our system, the expected temperature rise is minimal. Even for the case where we expect highest $\Delta T_S$ (while using 1 mM KCl solution with conductivity 0.93 mS/cm), the temperature rise within the system should be less than 1 °C. For the case where we used highest operating voltage (amplitude 3 V), the expected temperature rise is <0.05 °C.

## Discussion

Monolayer graphene can generate ultra-strong dielectrophoresis forces and outperform conventional metal electrodes that are orders of magnitude thicker. We have presented the concept of graphene-edge DEP and constructed a practical device platform to realize it. In our scheme, the critical dimension of graphene electrodes for DEP is controlled by its natural thickness and the insulator deposited by ALD, thus our approach is scalable and highly reproducible at the wafer scale. Using arrays of electrodes, we can selectively trap and position nanoparticles and DNA molecules precisely along the atomically sharp edges of graphene. This could facilitate performing of biological assays at low concentrations as well as exploring molecular interactions and conformational changes in confined space. We also demonstrated trapping of DNA molecules from 10 pM solution and on-chip assembly and precise positioning of nanodiamonds. Notably all of these rapid DEP manipulation steps were performed using monolayer graphene around or below 1 V, which makes the low-power graphene-edge DEP attractive for many applications. For example, edge trapping of biomolecules can be readily integrated with graphene nanoresonators[11, 40, 41] or tapered nanotips[7, 8] wherein field hotspots are located along the edges, enabling tunable mid-IR spectroscopy of ultralow-concentration molecules using graphene plasmons. Our wafer-scale chip platform can also be used for nano-positioning quantum emitters to build nano-photonic circuits or single-photon source arrays[42]. While we used straight edges of graphene in this work, DEP can also be performed with conductive nanopores[43, 44], presenting new opportunities for single DNA translocation and analysis[45–48]. Beyond graphene, atomically sharp edges of other 2D materials[49] can also be utilized for ultra-strong DEP, providing a practical route to building tunable biosensors.

## Methods

**Device fabrication**. Our graphene-edge DEP chip was fabricated starting from a thick layer of SiO₂ (300 nm) thermally grown on Si substrate. The gate electrode patterns were exposed and developed using photolithography, followed by a combination of reactive ion etching and wet etching on the SiO₂ layer. A metallization layer of Ti/Pd (10/40 nm) was then deposited by electron-beam evaporation and lifted off. Atomic layer deposition of HfO₂ at 300 °C was applied to form a dielectric layer (8 nm), which was annealed in nitrogen at 400 °C for 5 minutes. A via layer was patterned using photolithography, and then the HfO₂ in localized regions above the Pd metal was reactive-ion etched in SF₆ for 30–60 s. The via window through the dielectric layer is intended to provide the ability to subsequently contact the gate electrode with additional contact metallization. Single-layer graphene grown by chemical vapor deposition (CVD) was transferred onto the dielectric layer using a wet transfer process, patterned by photolithography, and then etched using an O₂ plasma. Ohmic contacts were patterned next using photolithography, and Cr/Au (10/80 nm) was deposited using electron-beam evaporation and lifted off. Finally, for the convenience of the device measurement, a thick metal layer of Cr/Al (10/300 nm) was patterned and deposited to form low-resistance electrical leads from the gate and Ohmic contacts to the contact pads for electrical probing.

**DEP experiments**. A solution volume of ~10 µl was placed on top of graphene electrodes, confined within a reservoir made in adhesive tape. A cover slip was placed to avoid any unwanted evaporation. A probe station was used to apply an AC bias across the contact pads by a function generator (HP 33120 A). The peak amplitude of the voltage used was in the range of 1–3 V. Depending on the particle, the frequency of the AC bias was optimized for polystyrene, DNA molecules, and nanodiamonds. For the polystyrene and ND particles, water was used as a surrounding medium (conductivity ~4 µS/cm), whereas 10 µM KCl solution (conductivity ~12 µS/cm) was used for DNA.

**Fluorescence microscopy**. Fluorescent nanoparticles used in this work are 190 nm diameter carboxylate-modified polystyrene beads ($\lambda_{ex}$: 470 nm, $\lambda_{em}$: 525 nm, Bangs Laboratories); high-pressure, high-temperature monocrystalline and carboxylate-modified nanodiamonds of type lb containing 10–15 NV centers, with average volumetric size of 40 nm ($\lambda_{em}$: 637 nm, Adamas Nanotechnologies); and YOYO-1 labeled DNA molecules of length 10 kbp and 500 bp ($\lambda_{ex}$: 491 nm). A laser driven light source (Energetiq) was illuminated through appropriate filters to excite fluorescence and the emitted light from the solution was collected through another filter. A 50× objective was used to observe nanoparticles and DNA molecules while using an image collection time of 200 ms for 190 nm beads, 1 s for the nanodiamond particles, 500 ms for the 10 kbp DNA and 1 s for the 500 bp DNA molecules. Fluorescent images were collected at regular intervals using a Photometrics CoolSNAP HQ2 CCD camera and Micro-Manager software, and they were further analyzed using ImageJ software.

**Data availability**. The data that supports the finding of this study are available from the corresponding author upon request.

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

## Acknowledgements

This work was supported by the National Science Foundation (NSF ECCS No. 1610333 to S.-H.O.); The Minnesota Partnership for Biotechnology and Medical Genomics (A.B., S.-H.O., Y.Z., and S.J.K.); J.B.E. acknowledges support from the EPSRC and ERC (Starter and Consolidator grants). A.B. acknowledges support from the University of Minnesota Doctoral Dissertation Fellowship. Device fabrication was performed at the University of Minnesota Nanofabrication Center, which receives support from the NSF through the National Nanotechnology Coordinated Infrastructure (NNCI) program, and the Characterization Facility, which has received capital equipment funding from the NSF through the MRSEC program under award no. DMR-1420013. Computational modeling was carried out using software provided by the University of Minnesota Supercomputing Institute. The authors thank Jonah Shaver for helping with the optics setup and Seon Namgung for helping with sample preparation.

## Author contributions

A.B. performed DEP trapping experiments. Y.Z. and S.J.K. performed chip design and fabrication. R.G. and T.L. performed theoretical calculations. B.P.N. and J.B.E. prepared DNA samples. S.-H.O. conceived and supervised the research. All the authors contributed to the analysis of the data and writing of the manuscript.

## Additional information

**Competing interests:** The authors declare no competing financial interests.

