## [Peer Review File · Nature Communications]

Reviewers' comments:

Reviewer #1 (Remarks to the Author):

In this paper the Authors describe experiments of dielectrophoretic (DEP) trapping for different nanomaterials (polystyrene beads, nanodiamonds, DNA molecules) within a novel two-dimensional (surface) scheme based on high-field gradients created by graphene sharp edges. The atomically thick graphene layer helps in increasing the field gradients and, consequently, DEP forces with respect to more standard metal electrodes. Through a careful design the Authors show DEP trapping of several nanomaterials and study the conditions of operation of the DEP forces (trapping, repulsion, minimum trapping voltage, etc).

This work falls within the on-going interest in on-chip devices that can trap and manipulate particles over a two-dimensional platform. The use of graphene presents an original route for increasing DEP forces and can open perspectives for increased trapping and manipulation at the nanoscale. The results are novel, the manuscript is well written and generally clear. The conclusions look sufficiently supported by the experiments. I only have some issues that the Authors should consider prior publication.

1. Introduction & Context.

The Authors outline the ongoing context for on-chip manipulation. For the case of optical trapping (plasmonic tweezers), in order to give a broader context, I suggest to add also a more general reference (e.g., Marago, et al., *Nature Nanotech* 8 (2013): 807-819; Juan, et al. *Nature Photon* 5 (2011): 349-356) since ref 16 seems to be very specific.

Moreover, among the different technique I would suggest to add also "optoelectronic tweezers" that combines the use of dynamic electric fields and light to move/trap particles (see, e.g., Wu, M. C. "Optoelectronic tweezers." *Nature Photonics* 5 (2011):322-324; Zhang, et al. *Manipulating and assembling metallic beads with Optoelectronic Tweezers. Sci. Rep.* 6, 32840).

2. Dipole approximation.

The Authors discuss DEP forces in the context of dipole approximation (eq. 1). Are there limitations for the applicability of the dipole approximation when the field gradients change so sharply in space? Since the fields are increased by the sharp graphene edges I would expect that using a multipolar approach to the problem would need to be considered for a better modelling of the experiments. Can the Authors comment on this issue?

3. In Eq. (8) the force is proportional to the real part of the polarizability, $\text{Re}\{\alpha\}$, rather than the polarizability, α . Despite the fact that in this specific case (DEP on DNA molecules) the polarizability has a negligible imaginary part, I think it is better to write the equation in its general form to be consistent with Eq. 1.

4. Thermal effects.

In many on-chip manipulation techniques thermal effects induced by dissipation of the fields are an issue. They can be either a detrimental effect towards trapping or, in certain cases, they can be exploited for thermophoretic manipulation (like in ref 21). What is the situation for the graphene platform described here? Are thermal effects to be expected for higher voltages or this is not an issue because dissipation is negligible? A comment on this issue would be of interest to the community.

5. The Authors discuss the advantage of this method also in terms of scalability. What are the realistic potential for the scalability of the technique? In particular what is the minimum distance between two traps that could be designed and hence the maximum number density of traps on a chip?

Reviewer #2 (Remarks to the Author):

Review

The manuscript describes a study, which uses a clever architecture to generate strong electrical field gradients at the corner/edge of graphene strips, using the tungsten gate as a planar electrode. These are used to trap different particles. The cornerstone of the work is the use of the edge of graphene to produce high gradients at low voltages due to their sharpness.

While the idea of graphene electrode driven DEP is not entirely new (despite the authors claims), I believe that their architecture has several important advantages compared to previous works: the electrode separation is well-controlled down to nm-scale via commonly available and fully scalable ALD processes, the sharp tip vs plane is close to the classic DEP geometry for creating the largest possible field gradient (with the W electrode playing the role as the large electrode), the structure is perfectly possible to upscale and works well.

So, as the works by Xie (see below) are largely overlooked (and not at all of same quality), I think this work could revitalize the idea of DEP, and provide a straightforward, effective way to implement high quality DEP trapping devices on large scale, which should be easy to incorporate in lab-on-chip and microfluidic devices.

Also I appreciate that the electrical field distribution is correctly calculated; sometimes (often) graphene is modelled as a thin metallic plate. The theoretical parts are well done and provides insights into the function and mechanisms involved.

The manuscript does not fully convince that graphene is making such an enormous difference compared to thin metal electrodes, and this is perhaps its weakest point.

Overall I find that the manuscript interesting and worthy of publication in Nat. Comm., provided the authors are able to answer my requests.

Comments

1. Graphene edges for dielectrophoretic assembly is not new. Xie, *Micromachines* 2015, 6, 1317-1330 and Xie et al, 10.1109/NEMS.2015.7147434 use graphene edge dielectrophoresis to assemble carbon nanotubes FETs. The authors should cite this, and explain the novelty in relation to these works.

2. The authors should explain better the benefits of using graphene edge DEP. In the introduction, the authors state that patterned graphene nanostructure DEP is highly desired by non-existent. It is nearly non-existent, but why is it highly desired, and for what purpose exactly?

3. The trapping volume of Graphene edge DEP is compared to CNTs, and the authors state the trapping of CNTs is inherently smaller? Are the authors speaking about the "edge" of a nanotube to be the tip or the side?. The tip of a CNT should provide a significantly greater field enhancement factor and due to its point like shape also a large field gradient, as is known from CNT field emission displays. If the authors refer to the side of a SWCNT as the "edge" (which is not clear from the text) this is nearly as sharp as the edge of graphene (so, smaller field enhancement) but not necessarily provide a smaller gradient force.

4. Along (3), I do not understand the argument of the slow diffusion of the target particles for CNT compared to graphene – why should the target particles diffuse more slowly in the case of CNTs?. Please clarify.

5. Cf. Trapping volume. It appears from fig 2d that the gradient force is a factor of 4 larger for the metal electrode compared to the graphene, until the particles are closer than 10 nm, below which the graphene wins. However, when particles have been pulled that far in, why does it matter that

the graphene edge is a little sharper? The figure very nicely shows that the metal edge have a larger trapping volume (due to the x4 larger gradient, which goes in Eq (1), and quite adequate field gradient below 10 nm. The difference in gradient force for the two structures is just an order of magnitude. To maintain the claim that graphene edges are far superior to metal edges (which can easily be made thinner than 20 nm) and provide "ultra-strong DEP", the authors should provide a more convincing argumentation, or moderate their claims, for me to accept this.

6. Overall, the authors claim superiority of the graphene edge DEP compared to an equivalent metal structure, but mostly through the use of adjectives such as "ultra" and "super". The difference between the 20 nm thin metal electrode and the graphene according to the calculations seems to be that the gradient force is significantly larger for the metal electrode, unless the molecules are nearly touching the electrode – and is sure to be trapped. I am not sure why the long range (From edge) electrostatic gradient is higher for the metal – but perhaps that is a consequence of the lower DOS in graphene. The authors should explain how the performance of two equivalent devices (one with metal strips and one with graphene strips) would compare, to the advantage of graphene edge DEP. The fastest and simplest way to do this is to fabricate the equivalent structures with metal strips and repeat (for instance) the experiment shown in Fig 3 to demonstrate that graphene edges gives any advantage. The authors should either compare with real devices, or – in a clear and transparent manner – refine their argumentation of graphene being better than metal electrodes in a real device.

A comment related to point 6 : to me, it may even be an advantage that the graphene edge provides a smaller, long range trapping (compared to the metal electrode) and a larger short-range trapping. DEP experiments can be difficult to control, i.e. often no material or way too much material is assembled between the electrodes. Perhaps the smaller, more localized (and very high) gradient helps to get the very neat and ordered DEP shown in the figures.

A minor point: several sentences in the abstract are quite vague and generic (i.e. the first). I think the work deserves are sharper, stronger abstract - but this is not a request, just a recommendation.

Reviewer #3 (Remarks to the Author):

In this study, the authors achieve dielectrophoretic manipulation of nanoparticles using nano-scale graphene electrodes applying signals with low potentials between 0.45 V to 3 V, which is usually far greater in a system with a more traditional electrode configuration.

The conclusions currently imply an overall improved performance in the dielectrophoretic manipulation of particles, when in reality it's a localized improvement at the corner edges of the graphene electrodes (when comparing with Figure 2d. Regardless, it is important to note that in this particular study, successful manipulation of 500 Kbp DNA or ~40 nm nanodiamonds is achieved using only a potential up to 3V, which enables the possibility of easy portability.

These results are of significance in the area of microscale electrokinetics of bio-nanoparticles, in a platform that enables dielectrophoretic manipulation applying low potentials that would reduce undesired electric-based phenomena that may negatively influencing the system.

There a few areas in which this manuscript could be improved. Presenting the results, there's no information regarding the number of repetitions for each of the evaluated scenarios, therefore there is no visual information regarding standard deviations or error bars in the corresponding plots. The intensity analysis performed enables the possibility of an appropriate statistical analysis on visual data such as microscope images.

Additionally, the simulation of the gradient of electric field square does not consider the signal frequency. The fact that the Hafnium Oxide is a dielectric which changes its insulating capacity based on the frequency of the applied signal [1] needs to be considered; the assumption of a fixed dielectric constant is misleading since at higher frequencies the material stop behaving as an insulator, influencing the electric field and therefore, the gradient of electric field square. Some of the experimental results were obtained using a 10 MHz frequency, suggesting a negative DEP scattering the particles away but it might be the case that there is no significant dielectrophoretic force present at all. Please improve the simulations by including the effect of the frequency of the signal in the dielectric material (using the complex permittivity at the different frequencies) and discuss how this affects the magnitude of the gradient of electric field square and the potential drawbacks and limitations when using this configuration.

A few more minor comments:

In Figure 2, consider adding simulation of the 20 nm electrode as well to compare spatial distribution of the electric field, and show the point at which the data is obtained for plot comparing them both.

In figure 4 is not clear whether all experiments were run with a frequency of 100 kHz.

In Figure 5 is not very clear by just looking at the image which sub-images correspond to the 10 Kbp DNA and which to the 500 Kbp, a more clear labeling could be helpful.

1 - Zhao, C., Zhao, C.Z., Werner, M., Taylor, S., Chalker, P. Dielectric relaxation of high-k oxides. *Nanoscale Research Letters*. 8:456,1-12 (2013).

Response to Reviewers – NCOMMS-17-14525 (Barik *et al.*)

We thank the reviewers for their very thoughtful reading of the manuscript. We found their comments and criticisms very valuable in further improving the manuscript. Below we discuss our changes. We include the reviewers' comments in italics and then our response.

Response to Reviewer #1

In this paper the Authors describe experiments of dielectrophoretic (DEP) trapping for different nanomaterials (polystyrene beads, nanodiamonds, DNA molecules) within a novel two-dimensional (surface) scheme based on high-field gradients created by graphene sharp edges. The atomically thick graphene layer helps in increasing the field gradients and, consequently, DEP forces with respect to more standard metal electrodes. Through a careful design the Authors show DEP trapping of several nanomaterials and study the conditions of operation of the DEP forces (trapping, repulsion, minimum trapping voltage, etc).

This work falls within the on-going interest in on-chip devices that can trap and manipulate particles over a two-dimensional platform. The use of graphene presents an original route for increasing DEP forces and can open perspectives for increased trapping and manipulation at the nanoscale. The results are novel, the manuscript is well written and generally clear. The conclusions look sufficiently supported by the experiments. I only have some issues that the Authors should consider prior publication.

[Response] We thank Reviewer 1 for carefully reading our manuscript and providing very helpful and encouraging comments. We have addressed all of the comments, as detailed below.

1. Introduction & Context.

The Authors outline the ongoing context for on-chip manipulation. For the case of optical trapping (plasmonic tweezers), in order to give a broader context, I suggest to add also a more general reference (e.g., Marago, et al., Nature Nanotech 8 (2013): 807-819; Juan, et al. Nature Photon 5 (2011): 349-356) since ref 16 seems to be very specific.

Moreover, among the different technique I would suggest to add also "optoelectronic tweezers" that combines the use of dynamic electric fields and light to move/trap particles (see, e.g., Wu, M. C. "Optoelectronic tweezers." Nature Photonics 5 (2011):322-324; Zhang, et al. Manipulating and assembling metallic beads with Optoelectronic Tweezers. Sci. Rep. 6, 32840).

[Response] We agree that these are more generally appealing references and have added all of the suggested references in the revised manuscript.

2. Dipole approximation.

The Authors discuss DEP forces in the context of dipole approximation (eq. 1). Are there limitations for the applicability of the dipole approximation when the field gradients change so sharply in space? Since the fields are increased by the sharp graphene edges I would expect that using a multipolar approach to the problem would need to be considered for a better modelling of the experiments. Can the Authors comment on this issue?

[Response] We agree with Reviewer 1 that a multipolar approach would be more accurate when gradient DEP forces vary significantly over the size of a particle. However, a dipole approximation in this case would still provide reliable numbers considering the fact that we are only trapping nanoscale

particles with size varying between 36 and 190 nm. To point out the limitations of our approach we have added the following texts to the manuscript.

“The time-averaged DEP force on a particle of radius R...For the case, where electric field gradient varies greatly over particle dimension, higher order moments (quadrupole, octopole, etc.) become important. However, in this case as our model system is based on nanoscale particles, we assumed a dipole approximation.”

3. *In Eq. (8) the force is proportional to the real part of the polarizability, $Re\{\alpha\}$, rather than the polarizability, α . Despite the fact that in this specific case (DEP on DNA molecules) the polarizability has a negligible imaginary part, I think it is better to write the equation in its general form to be consistent with Eq. 1.*

[Response] This is a good suggestion and we updated the text to specifically mention the real part of the polarizability is considered here:

“where α is the **real part of the** polarizability of the DNA molecule.²⁹”

However, the reason we used a different form of DEP equation (eq. 8) instead of the general form for spherical particles (eq. 1) is because for DNA molecules a counterion fluctuation model is used to predict its polarizability. The derivation for such is not as straightforward as for spherical particles where all the surface charges are uniformly distributed. We cited appropriate references for the polarizability values of the DNA molecules used in this work and also for the corresponding DEP equation (eq. 8).

4. *Thermal effects.*

In many on-chip manipulation techniques thermal effects induced by dissipation of the fields are an issue. They can be either a detrimental effect towards trapping or, in certain cases, they can be exploited for thermophoretic manipulation (like in ref 21). What is the situation for the graphene platform described here? Are thermal effects to be expected for higher voltages or this is not an issue because dissipation is negligible? A comment on this issue would be of interest to the community.

[Response] This is also a very good suggestion by Reviewer 1 and we have done further investigation to address this issue. As we used low operating voltages and low-conductivity solutions, the overall heating effects within the solution is minimal. We have added the following paragraph to the revised manuscript to address the heating issue.

“Stability of biomolecules is dependent on the temperature of the surrounding environment. We investigated the expected temperature rise in our system based on Joule heating from the relation $\Delta T_S \sim \sigma_m V^2 / 2k$ (where ΔT_S is the rise in solution temperature and k is the thermal conductivity). As we used low voltages and low-conductivity solutions in our system, the expected temperature rise is minimal. Even for the case where we expect highest ΔT_S (while using 1 mM KCl solution with conductivity 0.93 mS/cm), the temperature rise within the system should be less than 1 °C. For the case where we used highest operating voltage (amplitude 3 V), the expected temperature rise is less than 0.05 °C.”

5. *The Authors discuss the advantage of this method also in terms of scalability. What are the realistic potential for the scalability of the technique? In particular what is the minimum distance between two traps that could be designed and hence the maximum number density of traps on a chip?*

[Response] This is a very good suggestion to further emphasize the scalability of our device architecture, and we have added the following paragraph to the revised manuscript:

“The key feature of our processing scheme is the wafer-scale throughput and scalability of trap arrays. In the layout we employed, the gate finger spacing is 15 μm with two graphene segments on either side, and these segments have a pitch of 60 μm . The current design provides a density of 4 trapping sites per 900 μm^2 . This density can be further enhanced by reducing the pitch of the gate fingers as well as the graphene segments to 1 μm , which is realistic using i-line optical lithography. In such a case, the trapping site density could be increased to 4 sites per 1 μm^2 , a nearly 1000 \times improvement over the current devices. Furthermore, using advanced optical lithography or electron-beam lithography to form the gate fingers and segment the graphene, the trapping density could be further increased by another 1-2 orders of magnitude.”

Response to Reviewer 2

Recommendation: Publish after major revisions noted.

The manuscript describes a study, which uses a clever architecture to generate strong electrical field gradients at the corner/edge of graphene strips, using the tungsten gate as a planar electrode. These are used to trap different particles. The cornerstone of the work is the use of the edge of graphene to produce high gradients at low voltages due to their sharpness.

While the idea of graphene electrode driven DEP is not entirely new (despite the authors claims), I believe that their architecture has several important advantages compared to previous works: the electrode separation is well-controlled down to nm-scale via commonly available and fully scalable ALD processes, the sharp tip vs plane is close to the classic DEP geometry for creating the largest possible field gradient (with the W electrode playing the role as the large electrode), the structure is perfectly possible to upscale and works well.

So, as the works by Xie (see below) are largely overlooked (and not at all of same quality), I think this work could revitalize the idea of DEP, and provide a straightforward, effective way to implement high quality DEP trapping devices on large scale, which should be easy to incorporate in lab-on-chip and microfluidic devices.

Also I appreciate that the electrical field distribution is correctly calculated; sometimes (often) graphene is modelled as a thin metallic plate. The theoretical parts are well done and provides insights into the function and mechanisms involved.

The manuscript does not fully convince that graphene is making such an enormous difference compared to thin metal electrodes, and this is perhaps its weakest point.

Overall I find that the manuscript interesting and worthy of publication in Nat. Comm., provided the authors are able to answer my requests.

[Response] We thank Reviewer 2 for detailed and constructive suggestions, as well as his/her comments that our manuscript is worthy of publication in *Nat. Comm.* We have cited the paper by Xie *et al.* and addressed other concerns as detailed below. We hope that our revised manuscript is satisfactory for publication.

Comments

1. *Graphene edges for dielectrophoretic assembly is not new. Xie, Micromachines 2015, 6, 1317-1330 and Xie et al, 10.1109/NEMS.2015.7147434 use graphene edge dielectrophoresis to assemble carbon nanotubes FETs. The authors should cite this, and explain the novelty in relation to these works.*

[Response] We thank Reviewer 2 for pointing out Xie's paper wherein they used an AFM tip to mechanically cut a 94-nm gap in graphene and used the structure for trapping CNTs. Following Reviewer 2's suggestion, we have cited that paper as Ref. [32] in our revised manuscript. We also appreciate Reviewer 2 for positive comments recognizing that our structure is "*perfectly possible to upscale*" and "*straightforward, effective way to implement high quality DEP trapping devices on large scale, which should be easy to incorporate in lab-on-chip*". Indeed, we believe that the novelty of our architecture is the **wafer-level scalability**, the **precise and regular placement** of the trapped particles which arises due to our ability to pattern both the gate electrode and graphene, and the **low-voltage operation** which arises both due to the graphene and our use of a thin (8-nm) high-K gate dielectric which maximizes the DEP strength. This combination is essential for trapping small biomolecules (e.g. DNA used in our work), which have much lower polarizability than long CNTs used in Xie's work. They mentioned the possibility of using graphene to trap biomolecules but did not perform such experiments.

Our work is driven by the desire to trap biomolecules with low voltages along graphene edges and ultimately control the location of the trapping. We also demonstrate regular, precision edge trapping and nanopositioning of molecules at the junctions. In Xie's paper, such nano-positioning capability is not shown, as CNTs are dispersed over the whole electrodes at trapping voltages of ~10 V. As Reviewer 2 mentions later, one of challenges in DEP trapping is that "*often no material or way too much material is assembled between the electrodes.*"

We demonstrate low-voltage trapping and point toward how a new paradigm in sensing could be enabled by our scalable architecture. We hope that the inclusion of Ref. [32] by Xie *et al.*, along with these comments on low voltages and scalability, are satisfactory.

2. *The authors should explain better the benefits of using graphene edge DEP. In the introduction, the authors state that patterned graphene nanostructure DEP is highly desired by non-existent. It is nearly non-existent, but why is it highly desired, and for what purpose exactly?*

[Response] As we mentioned in the introduction, there is a lot of interest in using graphene ribbons for mid-infrared plasmonic biosensing (Ref. [11]) and graphene nanopores for single-molecule detection (Ref. 13). In such novel sensing applications of graphene, it is highly desirable to rapidly attract target biomolecules toward edges ("hot spots" for sensing) using low bias voltages.

3. *The trapping volume of Graphene edge DEP is compared to CNTs, and the authors state the trapping of CNTs is inherently smaller? Are the authors speaking about the "edge" of a nanotube to be the tip or the side?. The tip of a CNT should provide a significantly greater field enhancement factor and due to its point like shape also a large field gradient, as is known from CNT field emission displays. If the authors refer to the side of a SWCNT as the "edge" (which is not clear from the text) this is nearly as sharp as the edge of graphene (so, smaller field enhancement) but not necessarily provide a smaller gradient force.*

[Response] In principle, metallic SWCNTs – although not as sharp as graphene - can also act as "edge" as Reviewer 2 mentioned. In practice, creating a well-defined and dense wafer-scale array of long and metallic SWCNTs (plus individual electrical contacts on SWCNTs) is not as straightforward

as creating our graphene edge DEP platform.

4. *Along (3), I do not understand the argument of the slow diffusion of the target particles for CNT compared to graphene – why should the target particles diffuse more slowly in the case of CNTs?. Please clarify.*

[Response] Previous work on CNT-based DEP demonstrated trapping of particles around sharp tips where the field gradient is maximum. In that case, the trapping zone is smaller for a point-like CNT tip whereas a long graphene edge can generate 2D hemi-cylindrical trapping zones, thereby more efficiently and rapidly trapping molecules. If one could make a dense array of CNT tips that cover large areas, then CNTs would also work efficiently for large-scale trapping. However, making such a perfectly ordered array of CNTs is not as trivial as using graphene edges, and rapid low-voltage CNT-based DEP has not been shown via CNTs. We have noted this issue in the revised manuscript.

5. *Cf. Trapping volume. It appears from fig 2d that the gradient force is a factor of 4 larger for the metal electrode compared to the graphene, until the particles are closer than 10 nm, below which the graphene wins. However, when particles have been pulled that far in, why does it matter that the graphene edge is a little sharper? The figure very nicely shows that the metal edge has a larger trapping volume (due to the x4 larger gradient, which goes in Eq (1), and quite adequate field gradient below 10 nm. The difference in gradient force for the two structures is just an order of magnitude. To maintain the claim that graphene edges are far superior to metal edges (which can easily be made thinner than 20 nm) and provide “ultra-strong DEP”, the authors should provide a more convincing argumentation, or moderate their claims, for me to accept this.*

[Response] Please see our response to comment #6.

6. *Overall, the authors claim superiority of the graphene edge DEP compared to an equivalent metal structure, but mostly through the use of adjectives such as “ultra” and “super”. The difference between the 20 nm thin metal electrode and the graphene according to the calculations seems to be that the gradient force is significantly larger for the metal electrode, unless the molecules are nearly touching the electrode – and is sure to be trapped. I am not sure why the long range (From edge) electrostatic gradient is higher for the metal – but perhaps that is a consequence of the lower DOS in graphene. The authors should explain how the performance of two equivalent devices (one with metal strips and one with graphene strips) would compare, to the advantage of graphene edge DEP. The fastest and simplest way to do this is to fabricate the equivalent structures with metal strips and repeat (for instance) the experiment shown in Fig 3 to demonstrate that graphene edges gives any advantage. The authors should either compare with real devices, or – in a clear and transparent manner – refine their argumentation of graphene being better than metal electrodes in a real device.*

[Response] The gradient of the electric field depends on the radius of curvature of the electrode edge, which is much smaller for graphene (Angstrom scale) as compared to a realistic metal electrode. Thus the gradient of electric field is much stronger close to the graphene edge and it provides a larger short-range trapping capability. However, the long-range effect is weaker because of the atomic thickness of the graphene electrode – causing a faster decay in gradient field away from the electrode edge. We compared the performance of the graphene electrode to more realistic nanoslit electrode geometry, where a 50 nm gap was assumed between two 20-nm-thick metal electrodes.

Again in this case the long-range gradient field is stronger for the nanoslit geometry whereas the short-range gradient field is more than two orders of magnitude lower than the graphene edge. Hence it is clear that even if the short-range trapping efficiency is dependent on the radius of curvature of the electrode edge, the long-range effect in the bulk solution is dependent on the electrode geometry. Furthermore, we thank reviewer 2 for the comment on point 6 and we have added a few sentences to the manuscript to highlight the significance of the graphene electrodes.

“However, near the graphene edge, gradient forces are stronger than the metal electrode – demonstrating capability for larger short-range trapping. This in turn enables trapping of small number of analyte nanoparticles in a more controllable fashion at the edge of the graphene electrode without much interferences from the bulk solution.”

A comment related to point 6 : to me, it may even be an advantage that the graphene edge provides a smaller, long range trapping (compared to the metal electrode) and a larger short-range trapping. DEP experiments can be difficult to control, i.e. often no material or way too much material is assembled between the electrodes. Perhaps the smaller, more localized (and very high) gradient helps to get the very neat and ordered DEP shown in the figures.

[Response] We thank Reviewer 2 for these constructive suggestions. We agree that the more localized trapping force around graphene edges can benefit some applications. Besides, we could also consider stretching of molecules or causing conformational changes after trapping them along graphene edges. To reflect this comment, we have updated the manuscript accordingly, as detailed in our response to the previous comment.

A minor point: several sentences in the abstract are quite vague and generic (i.e. the first). I think the work deserves are sharper, stronger abstract - but this is not a request, just a recommendation.

[Response] Thank you for this kind suggestion, and we have revised the abstract accordingly.

Response to Reviewer 3

In this study, the authors achieve dielectrophoretic manipulation of nanoparticles using nano-scale graphene electrodes applying signals with low potentials between 0.45 V to 3 V, which is usually far greater in a system with a more traditional electrode configuration.

The conclusions currently imply an overall improved performance in the dielectrophoretic manipulation of particles, when in reality it's a localized improvement at the corner edges of the graphene electrodes (when comparing with Figure 2d. Regardless, it is important to note that in this particular study, successful manipulation of 500 Kbp DNA or ~40 nm nanodiamonds is achieved using only a potential up to 3V, which enables the possibility of easy portability.

These results are of significance in the area of microscale electrokinetics of bio-nanoparticles, in a platform that enables dielectrophoretic manipulation applying low potentials that would reduce undesired electric-based phenomena that may negatively influencing the system.

[Response] We thank reviewer 3 for positive and helpful comments. Following are our responses.

There a few areas in which this manuscript could be improved. Presenting the results, there's no information regarding the number of repetitions for each of the evaluated scenarios, therefore there is no visual information regarding standard deviations or error bars in the corresponding plots. The intensity analysis performed enables the possibility of an appropriate statistical analysis on visual data such as microscope images.

[Response] While more rigorous statistical analysis across several rounds of experiments would be great to present results in a quantitative way, our approach in this paper had been to show a proof-of-concept of the usability of graphene as an electrode for DEP manipulation for nanoparticles as well as biomolecules. We are able to present that by performing several experiments across multiple nanoparticles as well as biomolecules of different sizes at different concentrations. We performed experiments to show that these events are very much reproducible across several nanoparticles within a low voltage range. We tested the experimental setup for voltage dependence as well as frequency dependence for both polystyrene nanoparticles as well as DNA molecules.

Additionally, the simulation of the gradient of electric field square does not consider the signal frequency. The fact that the Hafnium Oxide is a dielectric which changes its insulating capacity based on the frequency of the applied signal [1] needs to be considered; the assumption of a fixed dielectric constant is misleading since at higher frequencies the material stop behaving as an insulator, influencing the electric field and therefore, the gradient of electric field square. Some of the experimental results were obtained using a 10 MHz frequency, suggesting a negative DEP scattering the particles away but it might be the case that there is no significant dielectrophoretic force present at all. Please improve the simulations by including the effect of the frequency of the signal in the dielectric material (using the complex permittivity at the different frequencies) and discuss how this affects the magnitude of the gradient of electric field square and the potential drawbacks and limitations when using this configuration.

[Response] The reviewer is correct that HfO₂ dielectrics can often display some degree of frequency dispersion as noted in [C. Zhao et al., "Dielectric relaxation of high-k oxides," *Nanoscale Res. Lett.* (2013) 8:456], which we cite as Ref. 35 in the revised manuscript. But we have found that this dispersion is a relatively weak effect over the frequency range considered in these experiments. The plot below shows the capacitance of an MIM capacitor where the HfO₂ was deposited using the same conditions as in the manuscript. The MIM capacitor consisted of a bottom Pd electrode, 7.5 nm of HfO₂ and a Cr/Au top electrode. The capacitor had an area of 4000 μm². As can be seen, the capacitance only decreased by ~ 10% between 1 kHz and 10 MHz, an effect we attribute to border traps in the HfO₂ [M. Ebrish, et al., *Dev. Res. Conf.*, 2013]. Based upon these experiments, it is clear

that the HfO₂ does not lose its insulating properties at the frequencies relevant to these experiments.

We do note, however, that the typical dielectric constant of our HfO₂ is well below the ideal value of 25, and has been found to be on the order of 13-17 in various MIM capacitors that we have measured using similar HfO₂ dielectrics, which suggests that further improvement in the capacitance scaling can be obtained by optimizing our ALD dielectric deposition conditions.

We have also updated our simulations to account for more realistic values of our HfO₂ dielectric constant. However, the effect of dielectric constant on the gradient force is negligible.

A few more minor comments:

In Figure 2, consider adding simulation of the 20 nm electrode as well to compare spatial distribution of the electric field, and show the point at which the data is obtained for plot comparing them both.

In figure 4 is not clear whether all experiments were run with a frequency of 100 kHz.

In Figure 5 is not very clear by just looking at the image which sub-images correspond to the 10 Kbp DNA and which to the 500 Kbp, a more clear labeling could be helpful.

[Response] We thank Reviewer 3 for these comments. We updated the figures in response. For figure 2, we noted the vertical cut lines on both 2a and 2c. However, we decided not to include the color map of the spatial field distribution, which looks very similar to the case of the graphene simulation as shown below. More appropriate quantitative comparison is already included in figure 2d.

In figure 4, the voltage dependence was performed at 1 MHz, which is currently noted in the figure as well as the caption.

In figure 5, we noted “10 kbp DNA” and “500 bp DNA” to avoid further confusions.

REVIEWERS' COMMENTS:

Reviewer #1 (Remarks to the Author):

I have considered the Authors rebuttal and revised manuscript and I feel that the points raised in the review process have been satisfactorily addressed by the Authors.

Reviewer #2 (Remarks to the Author):

The authors have provided adequate responses to the points I raised, and made amendments to the manuscript accordingly. I still feel that the manuscript is in some ways incremental - but perhaps just the paper the research field needs; there are some significant improvements over previous work, and overall the work is inspiring.

The ability to spark interest and generate research in an area with scientific and technological potential is there, and that, in my mind, justify publication in NC.

Reviewer #3 (Remarks to the Author):

After reading the revised manuscript, the authors were able to carefully address the main concerns regarding the behavior of the HfO₂ film over the range of frequencies employed in this study, and updated the value of dielectric constant employed in the simulations.

Additionally, if the study is considered a proof-of-concept of the DEP trapping performance of the nano-graphene electrodes over nano-particles with representative size range, the rigorous statistical analysis wouldn't be strictly necessary, although it remains under the final consideration of the editor.

Additional minor comments were also addressed allowing for an easier interpretation of the figures.